



# Evidence for pyrazine-based chromophores in cloudwater mimics containing methylglyoxal and ammonium sulfate

Lelia Nahid Hawkins[1], Hannah Greer Welsh[1], and Matthew Von Alexander[2]

[1]Dept of Chemistry, Harvey Mudd College, 301 Platt Blvd, Claremont, CA 91711
[2]Dept of Chemistry, Pomona College, Claremont, CA 91711

*Correspondence to:* Lelia N. Hawkins (lhawkins@g.hmc.edu)

**Abstract.** Simulating aqueous brown carbon (aqBrC) formation from small molecule amines and aldehydes in cloud water mimics provides insight into potential humic-like substance (HULIS) contributors and their effect on local and global aerosol radiative forcing. Previous work has shown that these (Maillard type) reactions generate products that are chemically, physically, and optically similar to atmospheric HULIS in many significant ways, including in their complexity. Despite numerous

characterization studies, attribution of the intense brown color of many aqBrC systems to specific compounds remains incomplete. In this work, we present evidence of novel pyrazine-based chromophores (PBC) in the product mixture of aqueous solutions containing methylglyoxal and ammonium sulfate. PBC observed here include 2,5-dimethyl pyrazine (DMP) and products of methylglyoxal addition to the pyrazine ring. This finding is significant as the literature of Maillard reactions in food chemistry tightly links the formation of pyrazine (and related compounds) to browning in foods. We investigated both

the roles of cloud processing (by bulk evaporation) and pH on absorptivity and product distribution in microliter samples to understand the contribution of these PBC to aqBrC properties. In agreement with previous work, we observed elevated absorptivity across the entire UV/visible spectrum following simulated cloud processing as well as higher absorptivity in more basic samples. Absorptivity of the pH 2 sample, following evaporation, exceeded that of the unevaporated pH 9 sample, indicating that cloud processing can overcome the previously observed kinetic barrier imposed on aqBrC formation in acidic conditions.

Further, the fraction of pyrazine compounds in the product mixture increased by up to a factor of four in response to drying with a maximum observed contribution of 16% at pH 5. Therefore, cloud processing under more acidic conditions may produce PBC at the expense of imine and imidazole-derived compounds. This finding has implications for further BrC reactivity and degradation pathways.



## 1 Introduction

Light absorbing organic carbon in atmospheric aerosol (brown carbon or BrC) has been shown to impact radiative forcing
with an estimated contribution of 19% to total aerosol absorption globally (Feng et al., 2013). AERONET measurements
over California have shown that in the brown carbon region (440 nm), brown carbon absorption is 40% of that attributed to

elemental carbon (Bahadur et al., 2012). Further, attribution of brown carbon absorption to elemental carbon (EC) can lead
to over-estimates in the predicted direct radiative forcing of EC, creating large model-measurement differences in aerosol
forcing and increasing the uncertainty in global climate models (Wang et al., 2014). Therefore, an accurate representation of
aerosol-climate interactions in models is not possible without correctly accounting for this ubiquitous material. Its formation
and persistence are dependent on BrC precursors making it highly variable and difficult to constrain. BrC has both primary

and secondary sources including biomass burning, fossil fuel combustion, and non-combustion biogenic emissions (Laskin
et al., 2015). Among the secondary sources are both gas phase and aqueous phase reactions. For example, nitrated aromatics
formed in gas phase reactions between aromatics and nitrogen oxides are a significant source of BrC in urban environments
like Los Angeles (Zhang et al., 2011). Aqueous phase reactions, in contrast, appear to form larger and highly functionalized
chromophores that often include reduced nitrogen species (Baduel et al., 2010; Duarte et al., 2004). This material can be formed

in Maillard-type reactions between small carbonyl compounds and ammonia or amines, which have been shown to produce
oligomeric species and BrC chromophores (termed aqueous brown carbon or aqBrC); a few representative studies include
De Haan et al. (2011), Kampf et al. (2012), Nguyen et al. (2012), Laskin et al. (2014), and Lin et al. (2015). Even simple
aqueous mixtures of one dicarbonyl (glyoxal or methylglyoxal) and ammonia or a small amine produces myriad products and,
under most conditions, yellow to brown color. In fact, BrC that is generated this way often resembles atmospheric HULIS

across a number of metrics, such as UV/visible absorption profile, mass spectral characteristics, and hygroscopicity (Hawkins
et al., 2016). Given that global SOA production from methylglyoxal and glyoxal alone account for an estimated glyoxal =
2.6 Tg C yr$^{-1}$ and megly = 8 Tg C yr$^{-1}$ (Fu 2008), and that 90% of their irreversible uptake occurs in clouds, Maillard type
reactions producing aqBrC could be an important global source of BrC.

Although careful work has suggested that organo-nitrogen species are largely responsible for the bulk absorptivity of these

amine-aldehyde systems (Bones et al., 2010), only a handful of aqBrC chromophores have been positively identified (Kampf et
al., 2012; Nguyen et al., 2012; Laskin et al., 2014; Lin et al., 2015; Aiona et al., 2017a). Most of these are functionalized
and conjugated imidazole products, summarized in detail in a recent review (Laskin et al., 2015). In many cases, the products
incorporate two or three nitrogen atoms through nucleophilic attack on a carbonyl by an amine or ammonia; the products often
grow through aldol condensation of methylglyoxal or glyoxal which may be acid (or base) catalyzed. The mechanistic details

of aqBrC formation, including the roles of pH and evaporation, are still incomplete, yet some trends in aqBrC absorptivity
have been identified. In most cases, it was concluded that chromophores form fastest in basic solution, because nitrogen (in
ammonia, methylamine, or amino acids) must initiate a nucleophilic attack at the carbonyl carbon atom (Yu et al., 2011; Kampf
et al., 2012). However, several studies have shown that aldol condensation of dicarbonyls can be acid catalyzed (Noziere et al.,
2009) and that aldol condensation products may contribute to visible light absorption (Sareen et al., 2010). This leaves the





relationship between pH and absorptivity in amine-aldehyde reactions unclear. For example, Sareen et al. (2010) showed that the peak absorbance (located at 282 nm) in the products of methylglyoxal/AS was linearly inversely related to the pH of samples after 24 hours, and concluded that the formation of light-absorbing products is enhanced by the presence of ammonium and hydronium ions through the acid-catalyzed aldol condensation mechanism. In a related study, Kampf et al. (2012) observed

a novel, highly absorptive chromophore in the glyoxal/ammonium sulfate system, the imidazole bicycle, whose production rate declined as the pH of the reaction dropped. Similarly, Yu et al. (2011) observed an exponential dependence on pH for the rate of formation of imidazole and related products and concluded that available ammonia, dictated by solution pH, drives this relationship. In the context of food chemistry, it is thought that neutral or slightly alkaline conditions favor melanoidins generally (Kwak et al., 2005) and in particular dimethylpyrazine (Koehler and Odell, 1970). Generally, it appears that basic

conditions favor chromophore formation for these reactions, limiting their potential for brown carbon formation in (slightly acidic) atmospheric water. Further, a number of these systems have been shown to generate acidic side products, such that even under favorable initial conditions, the potential for brown carbon formation appears negligible (Kampf et al., 2012). There may exist branching points in these reactions, favoring one type of chromophore over another according to the acidity. Noziere et al. (2009) observed that the iminium ion pathway, incorporating N, is faster at higher pH while the traditional aldol condensation

is favored at lower pH. This suggests that the incorporation of N-containing products has a pH dependence as well.

Previous studies have also established a correlation between aqBrC absorptivity and cloud processing (Nguyen et al., 2012; Powelson et al., 2013; Aiona et al., 2017b). Here cloud processing refers to the cycle of deliquescence to form cloud droplets, aqueous phase reactions, evaporation of those droplets to form residual particles, subsequent deliquescence, and so forth. Cloud processing was simulated in this and related work by dissolving reactants or aqueous extracts of SOA in water and then allowing

the solution to evaporate either in droplets or in bulk solutions. The absorptivity of the evaporated and redissolved solution in such simulations is significantly higher than the absorptivity of the mixed reagents that never undergo evaporation. Many of the proposed mechanisms for chromophore formation involve condensation (elimination of water), which is consistent with the observed browning. This seems to suggest that cloud processing increases BrC absorptivity irreversibly on the timescales studied. Evaporation during cloud processing also serves to increase reactant concentrations and decrease pH, though the

interplay of these effects is not understood. In particular, it has been shown that for laboratory generated limonene SOA, browning by evaporation is pH dependent (Nguyen et al., 2012). Under acidification to pH 2 with sulfuric acid, browning by organosulfate formation can occur during evaporation. Under mildly acidic conditions, the chromophores are thought to be imidazole-based. The absorption spectrum of the evaporated SOA material contains a stronger peak at 500 nm but lacks the shoulders at 430 nm and 570 nm observed under aqueous aging conditions. This suggests that evaporation serves mostly to

increase the rate of formation of the predominant chromophores (around 500 nm) but that some chemical differences exist between aqueous aging and cloud processed material. The authors suggest that evaporation may produce a narrower range of compounds than slower aqueous aging.

In this study we use chemical ionization mass spectrometry in concert with isotopically labelled ammonium sulfate to elucidate the roles of pH and evaporation in forming brown carbon chromophores from methylglyoxal and AS. Specifically,

we evaluate whether the increased absorptivity generated by evaporation is greater than the pH effect observed in previous





studies. Strong evidence is presented indicating that novel pyrazine-based chromophores form in this model system and that their contribution to total absorbance may be greatest under atmospherically relevant conditions.

## 2 Methods

### 2.1 Reagents and sample preparation

1 M stock solutions of each of the following reagents were prepared using low TOC, 18 MΩ resistivity water without further purification: 40 wt.% methylamine solution in water (Sigma Aldrich), 40 wt.% methylglyoxal solution in water (SAFC), glyoxal trimer dihydrate (Fluka Analytical), and ammonium sulfate (Sigma Aldrich). A 1 M solution of isotopically labeled ammonium sulfate $(^{15}NH_4)_2SO_4$ (Sigma Aldrich) was used to investigate the number of N atoms incorporated into products. The 18 MΩ resistivity water was verified as having less than 5 ppb total organic carbon by TOC analysis.

Each sample was prepared in a manner similar to De Haan et al. (2011) by combining 75 $\mu$L of 1 M ammonium sulfate (AS) or $^{15}$N AS with 75 $\mu$L of 1 M methylglyoxal (MG) in a 2-dram glass vial. An additional 45 uL of water, 1 M NaOH, or a combination was used to set the initial reaction pH at either 2, 5, 7 or 9. Vials were left to react in a dark hood. After 7 days, uncapped vials contained brown dried residual material and capped vials contain dilute, colored solutions.

### 2.2 Absorptivity

The solution mass absorption coefficient $MAC_{soln}$ as a function of wavelength was determined by UV/visible absorbance and total organic carbon (TOC) measurements following Hecobian et al. (2010) and Zhang et al. (2011, 2013). The spectroscopic setup includes a World Precision Instruments 3000 Series Liquid Waveguide Capillary Cell (LWCC-3100) with a 94 cm optical path length and an internal volume of 250 $\mu$L, an Ocean Optics lamp (DT-Mini-2), and an Ocean Optics detector (USB 4000-UV-Vis). A Sievers 5310C Laboratory total organic carbon (TOC) Analyzer was used to obtain TOC concentrations using 15%

ammonium persulfate (0.4 $\mu$L/minute flow rate) as the oxidizer and 6M phosphoric acid (1.0 $\mu$L/minute flow rate). The reacted samples were diluted with 100 mL - 1 L of water, depending on the brownness of the sample, in order to remain within the linear range of the Beer-Lambert relationship (measured absorbance less than 1 above 300 nm). Solutions were drawn through the waveguide and into the TOC at a rate of 0.5 mL min$^{-1}$ set by the TOC online sampling rate. Over the course of 30 minutes, 10 absorbance spectra and at least three TOC readings were collected for each discrete sample. These values were averaged

to obtain MAC bulk spectra shown in Figure 1. Water blanks were used between samples to avoid contamination and ensure readings return to nominally blank conditions. Eight samples were analyzed for solution MAC including both capped and dried samples at all four pH conditions.

Absorptivity in units of m$^2$ g OC$^{-1}$ is determined for each wavelength according to the following equation:

$$MAC_{soln} = \frac{A_\lambda - A_{ref}}{(0.94m)(TOC_{ppm})}log(10) \tag{1}$$





Where $A_{ref}$ is the absorbance in the reference region of 650-710 nm where no BrC absorbance is observed and is used to correct for changes in the total light transmittance due to movement of the fiber optic cables (Hecobian et al., 2010; Zhang et al., 2011, 2013).

## 2.3 Atmospheric pressure chemical ionization mass spectrometry

An Advion Compact Mass Spectrometer (CMS) with ASAP probe injection and atmospheric pressure chemical ionization (APCI) was used to obtain intact molecular ions (M+H$^+$) for both $^{14}$N and $^{15}$N containing samples. With the ASAP probe, both liquids and solids can be analyzed directly without extraction or even dissolution. The probe tip is dipped into the liquid sample or touched to the solid residue and inserted directly into the mass spectrometer. Conditions of ionization were nominally low temperature and low fragmentation meaning the capillary tube was at 135°C, the source gas was 250°C, and the capillary

voltage was 120V (source voltage held at 20V). The corona discharge was held at 5$\mu$A. A quadrupole mass spectrometer provides unit mass resolution spectra as the compounds are volatilized from the probe tip. 16 aqBrC samples were analyzed by APCI. For each pH condition studied (pH 2, 5, 7, and 9), four samples were prepared including capped and dried samples, with either $^{14}$N AS or $^{15}$N AS.

## 3 Results and Discussion

### 3.1 General absorbance and mass spectral characteristics

Figure 1 shows the measured absorptivity for the products of methylglyoxal/AS under all four pH starting conditions studied including pH 2, 5, 7, and 9 in both dried and capped samples (Figure 1a) as well as the log-log plot (representing the absorption Ångström exponent, AAE) for each sample (Figure 1b). All spectra are characterized by an absorption maximum at or near 300 nm with tails of varying intensity into the visible region. Samples that reacted without evaporation share a steep decline

in absorptivity after 325 nm and measurable absorbance up to about 425 nm; beyond 450 nm, no absorbance was observed. In contrast, the dried and redissolved samples possess measurable absorbance beyond 500 nm with clear shoulders at 350 nm. As summarized in Table 1, MAC$_{365}$ for the capped and dried samples ranges from 0.05-0.22 m$^2$g$^{-1}$ OC (capped) and from 0.59-1.13 m$^2$g$^{-1}$ OC (dried). These values compare well to water soluble OC from ambient measurements in the Bay of Bengal (0.2-1.5 m$^2$g$^{-1}$ at 365 nm by Srinivas and Sarin (2013)), in Dehli (2.7 m$^2$g$^{-1}$ at 300nm by Kirillova et al. (2013))

and in Chinese outflow observed in Gosan, Korea (0.8-1.1 m$^2$g$^{-1}$ at 365 nm by Kirillova et al. (2014)). Previous laboratory simulations of aqBrC formation report lower values of MAC$_{365}$ in the range of 100-500 cm$^2$g$^{-1}$ (Powelson et al., 2013) for these and similar mixtures. Absorption Ångström exponents in this study cover smaller ranges and show less dependence on pH than MAC$_{365}$, with AAE values from 9-12 for capped samples and from 7.7-8.9 for dried samples which are well within the range observed in Srinivas and Sarin (2013), Kirillova et al. (2013), and Kirillova et al. (2014). The role of evaporation in

altering absorptivity and generating products is discussed further in following sections.





Samples containing $^{14}$N AS and $^{15}$N-labeled AS were used pairwise for each reaction condition (pH and evaporation) to determine the number of N-atoms in each major product of the methylglyoxal/AS system. As is seen by comparing Figure 2a to Figure 2b and Figure 2c to Figure 2d, nearly all of the masses observed by APCI incorporate at least one N-atom and most incorporate two N-atoms. One product (m/z 162, not visible at pH 2) incorporates 3 N atoms; m/z 165 is barely visible

in Figure 2b. Only a few observed products, m/z 167, 199, and 271, show the same masses in both $^{14}$N and $^{15}$N samples. m/z 167 is consistent with a methylglyoxal self-reaction product reported in Sareen et al. (2010) as $C_6H_{15}O_5^+$ and another methylglyoxal self-reaction product in Lin et al. (2015) reported as $C_9H_{10}O_3^+$. Without a high resolution mass spectrum, we are unable to say which product is more likely forming in this study. However, the C9 compound was reported based on high resolution work in Lin et al. (2015) while the C6 product was based on 1-amu resolution data, therefore the evidence for the C9

compound is stronger. m/z 199 and 271 are also identified as MG self-reaction products. m/z 199 was assigned the molecular formula $C_{10}H_{14}O_4$ in Lin et al. (2015) though its structure remains undetermined; m/z 271 is very likely one MG addition to m/z 199. The masses observed in the pH 2 samples in Figure 2 compose most but not all of the products observed across all samples though the ratio of products is pH-dependent. APCI spectra for the remaining initial pH conditions (pH 5, 7, and 9) under both capped and dried conditions are provided in the Supplementary Material as figures S1-S3.

Many of the N-containing masses detected through APCI-MS have been observed previously in methlyglyoxal/AS reactions with electrospray ionization (De Haan et al., 2011; Lin et al., 2015; Aiona et al., 2017a) or aerosol chemical ionization with $H_3O^+$ and $I^-$ (Sareen et al., 2010) and their structures have been proposed with various levels of confidence in those studies. Table 2 provides masses and tentative structures for N-containing products observed here that are either identical to or consistent with those observed previously. These products can be grouped into one of two categories: imidazole-based compounds

(m/z 83, 97, 125, 197, 232, and 251) and linear imine-containing compounds (m/z 144, 180, 216, and 288). Our observations support the structural assignments in those studies; for example, we observed m/z 180 more often in dried samples and m/z 216 (completely hydrated form of m/z 180) only in pH 7 and 9 capped samples.

## 3.2 Pyrazine-based chromophores

In addition to imidazole and imine-containing products in Table 2, a series of products separated by 72 amu, beginning with m/z

109 and all containing 2 N-atoms, was observed in nearly all samples. m/z 109 was reported as a methylglyoxal/AS reaction product by electrospray ionization in Lin et al. (2015), without structural assignment but with a molecular formula of $C_6H_9N_2^+$ based on the exact mass. In addition, two products related to the delta-72 series by loss of 18 amu and still containing two N atoms were also observed with high frequency. Given the exact chemical formula of m/z 109 in Lin et al. (2015) and the evidence outlined below, we propose that m/z 109 is 2,5-dimethylpyrazine and that masses separated by 72 amu from m/z 109

are methylglyoxal addition products to 2,5-dimethylpyrazine (2,5-DMP) formed by aldol-type condensation reactions. The masses separated by 18 amu are also proposed as pyrazine-based structures, derived from dehydration. m/z 162 is proposed as an imine-substituted dehydration product of m/z 181 formed as shown in Scheme 3. Aromaticity of the heterocycle and frequent conjugation to the ring in these pyrazine products results in strong UV and (arguably) visible light absorption of the observed products; therefore, we will refer to these compounds as "pyrazine-based chromophores" (PBC) to distinguish



them from the chromophores listed in Table 2. Table 3 provides a complete list of proposed PBC, none of which have been previously reported with structural assignments in atmospheric chemistry studies. A number of these structures have been reported, however, in the chemistry of Maillard reactions in food and food models (Adams et al., 2008; Van Lancker et al., 2010; Yu et al., 2017; Divine et al., 2012).

5 The strongest evidence supporting 2,5-DMP as the correct structural interpretation of m/z 109 is the result of GC-MS analyses of ethyl acetate extracts of dried samples (e.g. dried pH 2, Figure S4). The residual brown material was diluted to 0.5 mL with ultrapure water containing 130 ppm pyrazine internal standard and adjusted to pH 9 with 1.0 M NaOH to encourage solubility in ethyl acetate. In Figure S4, the pyrazine internal standard is seen at 4.18 min RT and 2,5-DMP is visible at 6.57 min RT. The fragmentation pattern for the spectrum at 6.57 min is a 81% match to the NIST library spectrum for 2,5-DMP, a 13%

10 match to 2,6-DMP, and less than 5% match to a series of other pyrazine-based compounds. Figure S5 shows the comparison of the "Quick Search" result for the same peak, indicating a 91% match between the instrument's default library and the observed spectrum at 6.57 min. The same analysis was performed for dried samples at pH 5, 7, and 9. The peak area for 2,5-DMP in the gas chromatograms roughly corresponds to the pH-dependence of PBC measured by APCI described below, lending further support for this assignment.

15 There are three additional pieces of evidence pointing toward 2,5-DMP as the correct structure of m/z 109. First, the authors in Lin et al. (2015) speculated that the unidentified product $C_6H_8N_2$ must be aromatic because of its larger affinity for the biphenyl column than the similar compound, $C_6H_8ON_2$. Second, 2,5-DMP contains two intact methylglyoxal groups and has a plausible formation mechanism from our starting materials under atmospherically relevant conditions (Scheme 1). This mechanism is described as "the most accepted formation mechanism for pyrazine" in Maillard reactions in food (Adams et al.,

20 2008; Van Lancker et al., 2010) and is consistent with mechanisms presented in Van Lancker et al. (2010) (Scheme 1) and Yu et al. (2017). In addition, Divine et al. (2012) showed that among the four possible mechanisms of alkylpyrazine formation in aged Parmesan cheese, only the methylglyoxal-based mechanism correctly explains the observed products. Third, 2,5-DMP in particular has been shown to form in focused studies of Maillard reactions in food, specifically in cases where methylglyoxal was exposed to amine-containing compounds (Adams et al., 2008; Van Lancker et al., 2010; Divine et al., 2012; Yu et al.,

25 2017). In Van Lancker et al. (2010), the reaction of methylglyoxal with heated free amino acids and lysine containing dipeptides produced 2,5(6)-DMP and trimethylpyrazine at 10-100 fold times the signal of other pyrazines. Its presence in relatively large quantities suggests that the energetics of the 2,5-DMP pathway are favorable. However, our reactions contain ammonia instead of amino acids as in the Maillard reactions between sugar (or dicarbonyls) and amino acids in food. Therefore, a reducing agent is necessary to explain the formation of 2,5-DMP from our starting materials. Formic acid, especially under acidic conditions,

30 is a sufficiently strong reducing agent to complete the reductive amination step as shown in Scheme 1. Formic acid was not detected in this study in either positive or negative ion mode (as m/z 47 or 45, respectively) but it has been reported as a side product of glyoxal/AS and methyglyoxal/AS in several other studies (Galloway et al., 2009; De Haan et al., 2009; Sareen et al., 2010; Yu et al., 2011). The evidence outlined here lends confidence to our assignment of m/z 109 as 2,5-DMP.

 Observation of 2,5-DMP by GC-MS within our samples strengthens the structural assignment of related masses (m/z 162,

35 181, 235, 253, and 289) in Table 3, but direct evidence of these compounds remains lacking. However, there are again several





lines of evidence suggesting that methylglyoxal adds to 2,5-DMP in an aldol-type reaction as proposed in Table 3 and Scheme 2 and not in another configuration. First, m/z 162, 235, and 253 were reported in Lin et al. (2015) as $C_9H_{12}N_3^+$, $C_{12}H_{15}N_2O_3^+$, and $C_{12}H_{17}N_2O_4^+$, respectively, using HR mass spectrometry, in agreement with our proposed formulas. Second, these structures are based on the mechanism shown in Scheme 2 which follows the same acid-catalyzed aldol condensation mechanism

used to explain the previously identified imidazole derivatives in Table 2 and in De Haan et al. (2011) and Aiona et al. (2017a). Third, the relative abundance of these PBC in capped and dried samples follows logically from their structures. For example, m/z 235 and m/z 253 are related by loss of water and their relative abundance shifts in favor of m/z 235 in dried samples. m/z 162 likely forms through imine substitution and dehydration of m/z 181 as shown in Scheme 3. Imine substitution is favored in mildly acidic conditions like those used here and indeed m/z 162 is elevated in dried samples and not observed at pH 9. A

product with this molecular formula, $C_9H_{11}N_3$, was reported in Table 1 of Lin et al. (2015) with an exact mass of 162.10225, while the structure provided in their Supporting Information corresponds to a different formula, $C_6H_{11}O_4N$, matching the structure initially reported in Sareen et al. (2010). No structure was provided matching the observed ion $C_9H_{11}N_3$, however.

     While methylglyoxal could add to 2,5-DMP directly onto the aromatic ring (as in Adams et al. (2008) Scheme 1b), the resulting product mass m/z 165 ($C_9H_{13}N_2O^+$) did not rise above the baseline in our spectra. It was observed in Lin et al.

(2015); therefore, it is likely forming in our reactions as well, just below our detection limit. In Lin et al. (2015) the UV/visible absorption spectrum for m/z 165 shows significant absorption beyond 400 nm (peak 20 in Figure 3b), which is consistent with the structure in Adams et al. (2008). In fact, this acetylated pyrazine product is known to the flavor industry as having a yellow-brown color. Even though m/z 165 ($C_9H_{13}N_2O^+$) itself was not observed in this study, it is likely that the observed PBC contribute to visible light absorption. For example, m/z 253 (peak 14 in Lin et al. (2015) Figure 3b) has significant visible

light absorption detected following chromatographic separation. Further, the structural similarity between acetyl pyrazine and the products described here suggests that all of the methylglyoxal addition products should share some visible light absorption, particularly m/z 162, 235, and 289 since those compounds have carbonyl or imine groups conjugated to the pyrazine ring. Perhaps more convincing is the work presented in Yu et al. (2017) showing that pyrazines were strongly and positively correlated to browning products from ascorbic acid and amino acids as well as the results of Divine et al. (2012) connecting

browning in Parmesan cheese with total pyrazine production. Therefore, the formation of PBC from Maillard type reactions under atmospherically relevant conditions has implications for the radiative forcing potential of aqBrC SOA. Future LC-MS work targeting PBC is needed to assess the extent of the contribution of PBC to aqBrC.

     Reaction products of the methylglyoxal/AS system listed in Table 2 are very typical Maillard reaction products, observed in both cloud water simulations (De Haan et al., 2011; Lin et al., 2015; Sareen et al., 2010; Hawkins et al., 2016) and the chemical

characterization of baked bread, cooked meat, aged cheese and other reactions between carbonyl containing compounds and amines or amino acids (Koehler and Odell, 1970; Adams et al., 2008; Van Lancker et al., 2010; Divine et al., 2012). It is therefore surprising that pyrazine-based products, which are also well-known to form in Maillard reactions in food, have not been identified in the context of atmospheric reactions. One plausible reason that PBC were not observed in previous analyses is the strong acidity of the N-heteroatom in pyrazine compared with imidazole, leaving most PBC unprotonated in electrospray

solutions. For example, the $pK_a$ of pyrazine is 0.6 and 2,5-DMP is 1.6, whereas imidazole has a $pK_{a1}$ of 6.9 and a $pK_{a2}$ of





14.4. Unless the electrospray ionization solvent or column effluent (in LC-MS) was acidified below pH 2, there is no reason to expect significant quantities of pyrazine in its acidic, ionized form. Rather, predominant products observed in positive ion mode would be imidazole derivatives. Atmospheric pressure chemical ionization, however, does not require analytes to exist in their ionic form in the sample solution. Instead, the sample is gently ionized through a proton transfer from a hydronium

ion. Another reason for their apparent absence is the role that evaporation plays in forming pyrazine compounds. As detailed below, evaporation seems to drive PBC formation, consistent with findings from the food chemistry literature that high water content can inhibit pyrazine formation (Pletney, 2007). Although solutions were dried in De Haan et al. (2011), they were not dried in Lin et al. (2015) and only rapidly dried in Sareen et al. (2010) during atomization for aerosol CIMS analyses.

If drying and chemical ionization are necessary to observe the PBC in the methylglyoxal/AS system, they should have been

reported in Sareen et al. (2010) where atomized solutions were analyzed by aerosol CIMS. m/z 109, 181, and 235 were in fact observed in that study, but they were assigned to a water cluster and two MG self-reaction products in part due to their appearance in both AS and NaCl samples and to the absence of high resolution spectra necessary to constrain a molecular formula (0.5-amu resolution was used). m/z 181 was assigned to either $C_6H_{13}O_6^+$ or $C_6H_{11}O_5^+ \bullet H_2O$ and m/z 235 was assigned to $C_9H_{15}O_7^+$. However, the mass observed for m/z 181 (181.2 amu) is slightly closer to our proposed molecular

formula $C_9H_{13}N_2O_2^+$ at 181.21 amu than $C_6H_{13}O_6^+$ at 181.16 amu. Similarly, the exact mass for our proposed formula for m/z 235 ($C_{12}H_{15}N_2O_3^+$) is 235.26 which is closer to the observed value of 235.3 than the exact mass for $C_9H_{15}O_7^+$ (235.21). While inconclusive, it is possible that the PBC structures reported here were observed in Sareen et al. (2010) and if so, their abundance in that study is comparable to the abundance observed here since m/z 181 and 235 are both prominently featured in Figure 7 of Sareen et al. (2010). As to their presence in NaCl containing samples (where no ammonium was added), we can

only speculate that ammonia is notoriously difficult to avoid in many laboratory studies.

### 3.3    Role of pH in chromophore formation

To explore the role of pH in this work, we first compare the absorptivity of the capped samples after one week of reaction time. Figure 1a shows that absorptivity across the spectrum is positively correlated with pH. This can also be seen in Figures 3 and S6; Figure S6 is a photograph of sample vials taken 24 hours after adding methylglyoxal to AS. In our study, absorbance

has been normalized by TOC concentration for each sample to provide a bulk solution phase mass absorption coefficient, which accounts for the loss of any organic carbon to the gas phase. The results in Figure 1a are consistent with Kampf et al. (2012) and Yu et al. (2011) and suggest that over a pH range between 2 and 9, the role of nitrogen as a nucleophile is more important for chromophore formation than acid-catalyzed aldol condensation. The observed changes in absorptivity (rather than absorbance) imply that solution pH can promote one product (like methylglyoxal self-reaction products) at the expense

of another (N-containing products) or that unreacted methylglyoxal remains in solution after one week. However, none of the APCI spectra show peaks for unreacted methylglyoxal (m/z 73) or the n=1 water cluster addition (m/z 91). This suggests that either APCI is not favorable for detecting methylglyoxal or that methylglyoxal completely reacts with itself or ammonia (or sulfate) in one week. Therefore, the higher absorptivity observed at pH 9 suggests that stronger chromophores preferentially form under mildly basic conditions.





Figure 4 shows APCI spectra for the same four capped solutions after one week of reaction. The most consistently prominent ions are m/z 83 and 125, both imidazoles and both shown in Scheme 2 of De Haan et al. (2011). m/z 125 makes up more than 60% of the peak area in pH 2 and pH 5 samples. This ion was observed in Lin et al. (2015) and De Haan et al. (2011) and assigned to 4-methyl-2-acetyl imidazole (MAI) which is the methylglyoxal analog of imidazole carboxyaldehyde (IC) reported

in Galloway et al. (2009), Yu et al. (2011), and Kampf et al. (2012) using glyoxal instead of methylglyoxal. m/z 83 is 4-methyl imidazole (MI) generated by loss of a C2 group from the precursor to MAI. In the pH 7 samples, the m/z 125 contribution drops below 15% even though it is still the largest signal. At pH 9, m/z 125 is still about 15% of the peak area but the largest signal belongs to m/z 83 with 24% of peak area. These two major products appear to form in competition with one another, again consistent with Scheme 2 in De Haan et al. (2011). Only one pathway, however, promotes further oligomerization by aldol

condensation and that pathway forms m/z 125 and appears to be favored in the more acidic samples. A third major product in the basic samples is m/z 97 (dimethylimidazole) which is also shown in Scheme 2 of De Haan et al. (2011); we observe this ion under neutral and basic conditions when m/z 83 is prominent, consistent with that mechanism. Given that the pathway leading to further oligomerization by aldol condensation is favored under acidic conditions, it is curious that the basic samples possess higher absorptivity. This observation suggests that the previously reported imidazole oligomers formed by condensation are not

primarily responsible for the dark brown color of most basic methyglyoxal/AS solutions though they are likely to contribute some light absorption in all samples.

Overall, pH 2 and pH 5 capped samples do not display a large difference in the product distribution based on visual interpretation despite the fact that the pH 5 sample is clearly more absorptive. Looking closely, the PBC at m/z 181 increases from 2.5% of the signal to 3.7% and m/z 162 increases from 1.1% to 2.4% at pH 5, though using relative peak area from APCI data is

20 only semi-quantitative. These increases accompany a 10% drop in the contribution from m/z 125. At pH 7, two methylglyoxal self-reaction products become visible, m/z 199 and m/z 271, totalling 17.6% of the signal. Also at pH 7, dimethylimidazole appears. The largest PBC signal at pH 7 is m/z 253 at 3.2% (seen in the [15]N sample at m/z 255). Interestingly, at pH 9 the product distribution changes dramatically, and the largest signal is m/z 83 (MI) with dimethylimidazole (m/z 97) and MAI (m/z 125) close behind. One explanation for the dramatic shift in the product distribution is that the $pK_a$ of ammonium is 9.24 and

25 at pH 9, nearly 40% of the ammonium is present as ammonia while at pH 7, the fraction is less than 1%. Neither methylimidazole nor dimethylimidazole are known to have significant visible light absorbance, yet the pH 9 solution has the highest absorptivity. Over 80% of the peak area in the pH 9 sample can be accounted for without including any known (or likely) visible light-absorbing products. Because the absorptivity metric accounts for OC concentration, we cannot simply attribute the higher absorptivity to higher a concentration of chromophores driven by the faster reaction rate under basic conditions.

Rather, this result supports the previously proposed idea that a few highly absorbing species, present at low concentrations, are driving the overall absorption (Nguyen et al., 2012). For example, m/z 126 (1.3% of signal) is the dehydration product of m/z 144 (Table 2) and has a series of four conjugated double bonds. Similarly, m/z 180 (0.9% of signal) has six conjugated double bonds. However, these two particular compounds are present at higher relative concentrations in the pH 7 sample than at pH 9, which means that either the pH 9 sample contains additional strong chromophores or that the pH 7 sample contains some

non-absorbing products absent at pH 9.





Figure 5 illustrates the comparison among the pH 2-9 capped samples over the high mass region. Intensity is calculated relative to the single largest signal and is therefore not a useful metric for comparing concentrations across the four samples shown. However, it is clear that the pH 2 and 5 solutions have more PBC (m/z 181 and m/z 162) than other high mass products while at pH 7, the MG self reaction products at m/z 199 and m/z 271 are the predominant products. At pH 7-9, the contribution

of masses greater than m/z 200 increases, in particular, m/z 234 and 288 (Table 2). m/z 234 is the triple dehydration product of m/z 288, a linear imine-containing aldol condensation product. Once dehydrated, this product contains a series of 8 conjugated pi systems giving it potential to absorb well into the visible region. The percent area attributed to m/z 234 and m/z 288 increases from undetectable at pH 2 and 5 to 1-2% at pH 7-9. Although the pH 7 sample has a larger contribution from m/z 234 and 288, it also has a very large contribution from non-absorbers m/z 199 and 271. Together those two products make up a larger fraction

of signal than any other product observed in the pH 7 spectrum (17.6%). Further quantitative characterization is necessary before attributing specific products to the pH-dependent absorptivity observed here, but oligomers with masses beyond m/z 200 may be responsible for the observed absorptivity of these solutions.

With respect to PBC only, a more unique pH-dependence was observed. The contribution of all observed PBC to the total signal as a function of pH and evaporation is shown in Figure 6. We observed an increase in PBC contribution from 4.4%

to 7.4% from pH 2 to 7, with a dramatic drop to 2.2% at pH 9 (contrasting the more linear pH dependence of absorptivity). Mildly acidic conditions should favor PBC given the proposed mechanism in Scheme 1 since reductive amination is favored with some acidity, but highly acidic solutions prevent nucleophilic attack by ammonia on the aldehyde. While absorptivity on the whole increases with pH, the contribution from PBC to brown carbon absorbance is greatest under acidic conditions. The role of pH in the dried samples is discussed further in the next section where the effect of evaporation is considered.

**3.4 Role of evaporation in chromophore formation**

Figure 1a shows that under the range of acidities studied, evaporation increased the absorptivity across the entire UV/visible spectrum and generated a noticeable shoulder around 350 nm with a tail that extends beyond 550 nm. In the capped samples, the tail extends only to 425 nm. In previous comparisons, absorbance spectra from dried and redissolved material displayed narrower peaks than the slowly aged samples (Nguyen et al., 2012). This difference was attributed to the time allowed for

reaction, with the assumption that over short reaction times, the number of different products forming may be limited, creating narrow peaks in the absorbance spectra. However, the dried samples in this work reacted for the same length of time as the capped samples, and at the same temperature. This may explain why the peaks in absorbance spectra for our dried samples are as broad as the capped (aqueous aging) samples. As in Nguyen et al. (2012), the brown residual material was stable with respect to hydrolysis for for at least several days. The pH dependence of absorptivity in the dried samples mirrors the capped,

with the pH 9 dried sample being the most absorptive. Evaporation does not appear to change the relationship between pH and absorptivity. In fact, in both the aqueous and dried samples, the pH dependence of absorptivity between 350 and 400 nm is nearly linear (Figure 3), though the dried samples show a stronger dependence on pH than the capped samples. One of the most significant findings of this work is that the pH 2 dried sample was more absorptive than the pH 9 capped sample (by 0.4 $m^2g^{-1}$ over the 350-400 nm range), suggesting that cloud processing may overcome any barriers to amine-aldehyde browning



created by cloud water acidity. That is, evaporating acidic cloud droplets allows browning beyond that observed in even the most basic conditions. This effect is also visible in the AAE illustrated in Figure 1b. A larger difference exists between dried and capped samples, rather than acidic and basic ones. Given the number of water elimination steps in forming the products described here and in related studies, the observed effect of evaporation is unsurprising. The extent to which evaporation drives

browning in all amine-aldehyde pairs remains to be seen.

Figure 7 shows the APCI spectra of the residual material obtained from dried samples for pH 2-9. As with the capped samples, the contribution from m/z 125 decreases with pH while the C2 loss pathway in De Haan et al. (2011) (m/z 83 and 97) is increasingly favored with higher pH. PBC appear only in the acidic and neutral samples, and m/z 97 appears only in the neutral and basic samples. Unlike in the capped samples, however, the methylglyoxal self reaction products (m/z 199 and 271)

are nearly gone. One exception is m/z 167 which appears in the baseline of the pH 2 sample. The differences between capped and dry samples are better observed in Figure 8 where the capped sample spectrum for each pH is used as the background signal for the dried sample spectra. In both acidic samples we see a new peak, m/z 272, an unknown product with one nitrogen atom. At pH 2, m/z 181 contributes over 10% of the observed signal compared with 2.5% in the capped sample. At pH 5, most of the PBC are prominently visible, including m/z 109, 162, 181, 235, and 253 totalling 16% of signal (Figure 6). Contribution

of PBC to the total signal in the dried pH 5 sample is noteworthy, since this sample is arguably the most similar to ambient cloud processed secondary aerosol material. At pH 7, the PBC contribution drops to 10.1%; although that still places the pH 7 sample well above any of the capped samples, with a maximum PBC contribution of 7.4%. At pH 9, no PBC were observed in the dried sample. Another interesting trend visible only in the dried samples is the relative contribution of m/z 162 and m/z 181 as the predominant PBC species. At pH 2, m/z 162 is not detected while m/z 181 is just over 10% of total signal, at pH 5

the ratio of m/z 162:181 is 0.8 and at pH 7, the ratio is about 1.6 with m/z 162 making up 3.7% of signal. Neither was detected at pH 9. m/z 162 is the imine-substituted and dehydrated version of m/z 181 (Scheme 3) and the necessary substitution may be favored under the higher ammonia concentration of the neutral solution.

## 4    Conclusions

This study provides compelling evidence for the presence of novel pyrazine-based chromophores in the product mixture re-

sulting from aqueous Maillard-type reactions between methylglyoxal and ammonium sulfate. The presence of these PBC has not previously been reported in atmospheric chemistry studies, although their formation from these starting materials and subsequent browning effects have been widely observed in Maillard-type reactions in food chemistry. Both the absorptivity and the relative abundance of the various PBC showed a clear dependence on sample pH. Absorptivity was greatest under basic conditions and comparable to ambient measurements while PBC had the most significant contribution to products detected

by APCI (up to 16%) in samples at or below pH 5. The appearance of PBC in acidic and dried samples indicates that PBC's contribution to absorbance is likely non-negligible in cloud, fog, and aerosol water. It is notable that while absorptivity showed a positive linear dependence on sample pH, evaporation was shown to overcome the barrier to aqBrC formation imposed by acidic conditions. Evaporation of even the most acidic sample resulted in an array of products with higher average absorptivity




than the most basic unevaporated sample. Future work is needed to quantify the respective contributions of PBC and other chromophores to absorbance using chromatography coupled to mass spectrometry, to confirm the structural assignments proposed in this work, and to identify PBC in other related systems as well as in atmospheric samples. Further, these compounds are less likely to be observed by electrospray-based analyses because of their acidity so their formation and loss under atmospheric conditions and in atmospheric samples needs to be quantified in targeted studies.

*Code availability.*

*Data availability.* APCI and absorption spectra are publicly available as text files at
http://www.hmc.edu/hawkinslab/LeliaHawkins_HMC/Research/Research.html

*Code and data availability.*

*Author contributions.*

*Competing interests.* The authors declare that they have no competing financial interests.

*Disclaimer.*

*Acknowledgements.* The authors would like to acknowledge Prof. David Vosburg for his intellectual contribution to the pyrazine formation mechanisms. L.N. Hawkins was supported by NSF AGS-1555003, Research Corporation Cottrell College Award 22473, and the Barbara Stokes Dewey Foundation. H.G. Welsh was supported by the Kubota Fellows Program of the Harris Family Research Fund in Chemistry.





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




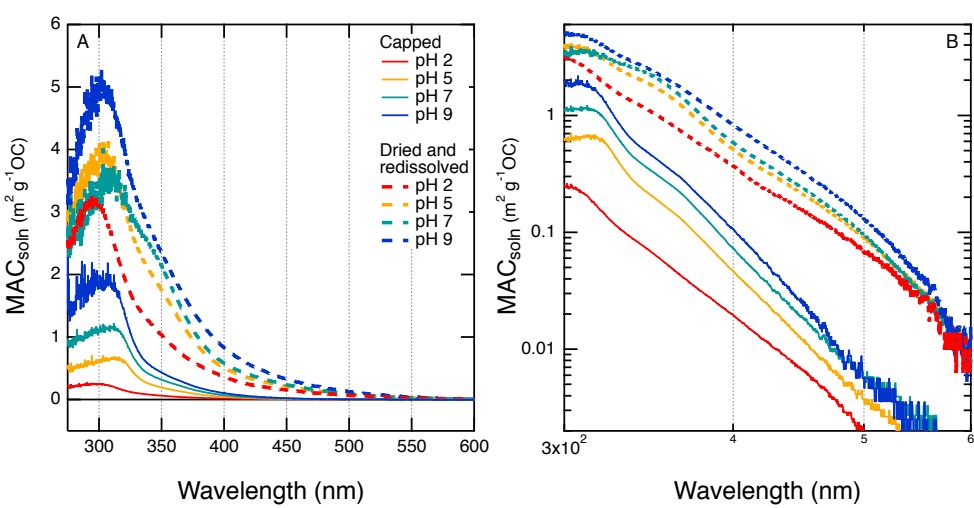

**Figure 1.** Normalized UV/visible absorbance spectra (displayed as mass absorption coefficient) on linear (a) and log (b) scales. Dashed lines indicate samples that were evaporated to dryness and redissolved prior to measurement. Spectra have been baseline corrected using any measured absorbance between 650 and 710 nm.



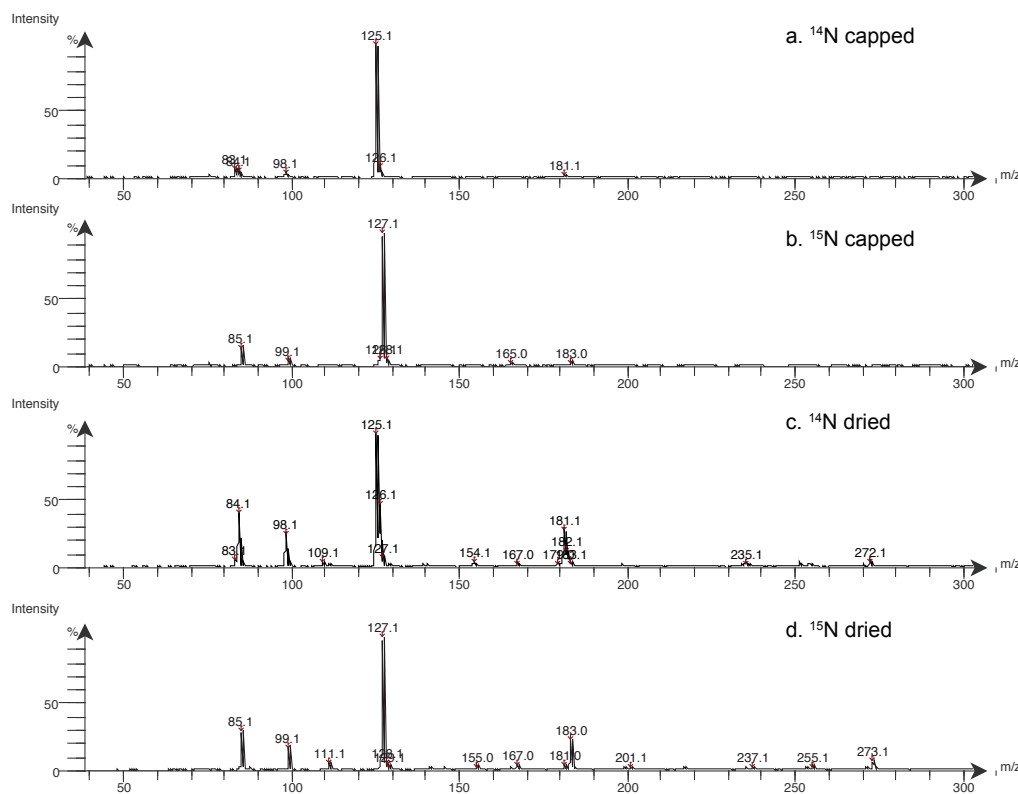

**Figure 2.** Atmospheric pressure chemical ionization (APCI) mass spectra for all four types of pH 2 samples including those with and without isotopically labeled N. Intensity is set to 100% for the largest signal and does not correspond to concentration in the sample itself.





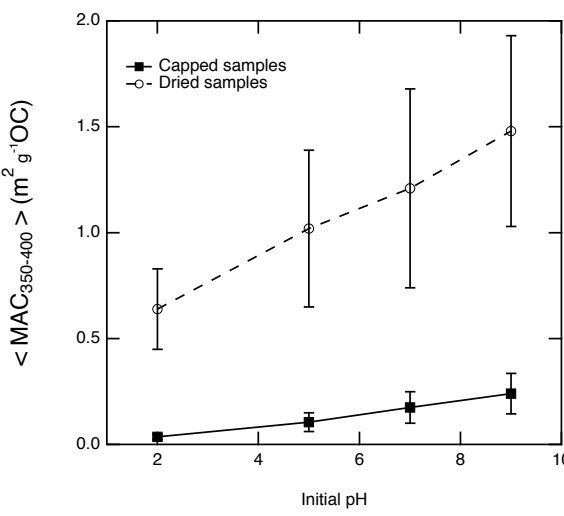

**Figure 3.** Solution-based mass absorption coefficients (MAC) calculated from 350-400 nm as a function of initial solution pH for both capped and dried samples. While both capped and dried samples become more absorptive in basic conditions, the pH dependence is more severe following evaporation to dryness.





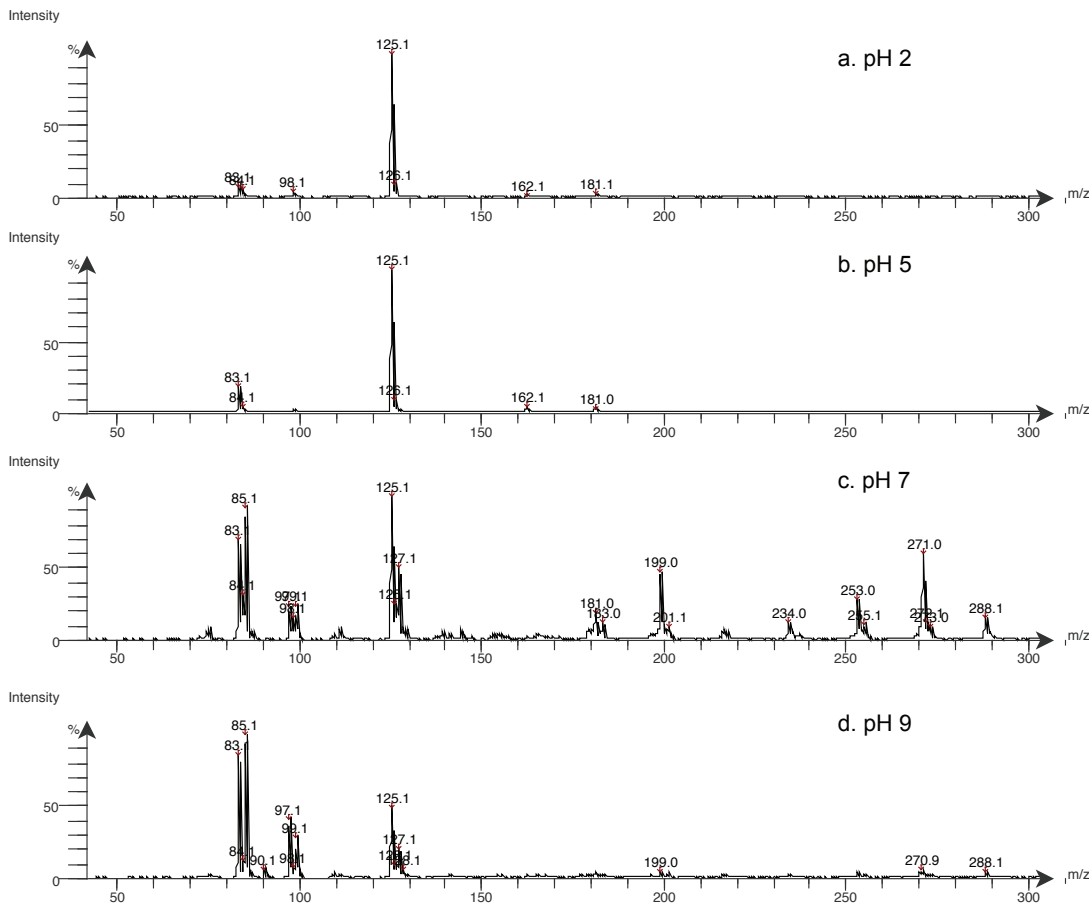

**Figure 4.** Atmospheric pressure chemical ionization (APCI) mass spectra for all four initial pH conditions in capped samples. Intensity is set to 100% for the largest signal and does not correspond to concentration in the sample itself.





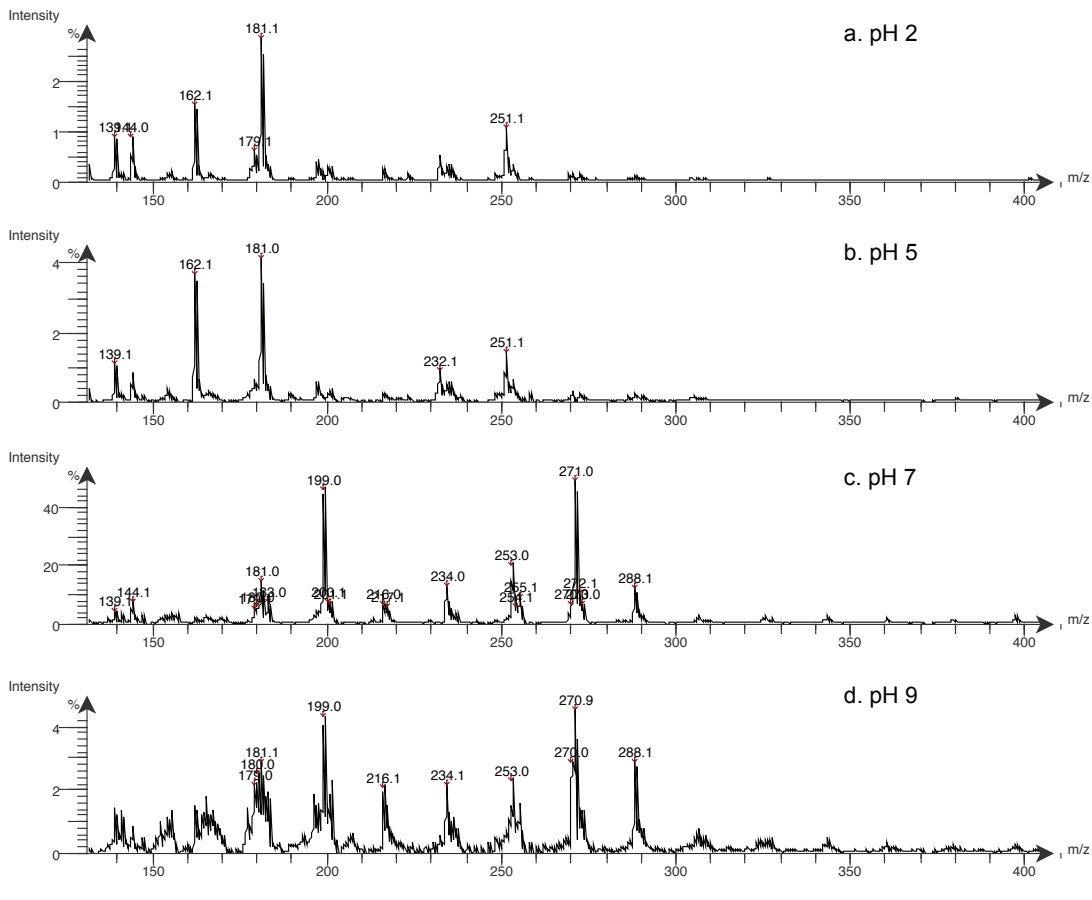

**Figure 5.** Atmospheric pressure chemical ionization (APCI) mass spectra for m/z greater than 130 for all four initial pH conditions in capped samples. Intensity is set to 100% for the largest signal and does not correspond to concentration in the sample itself.





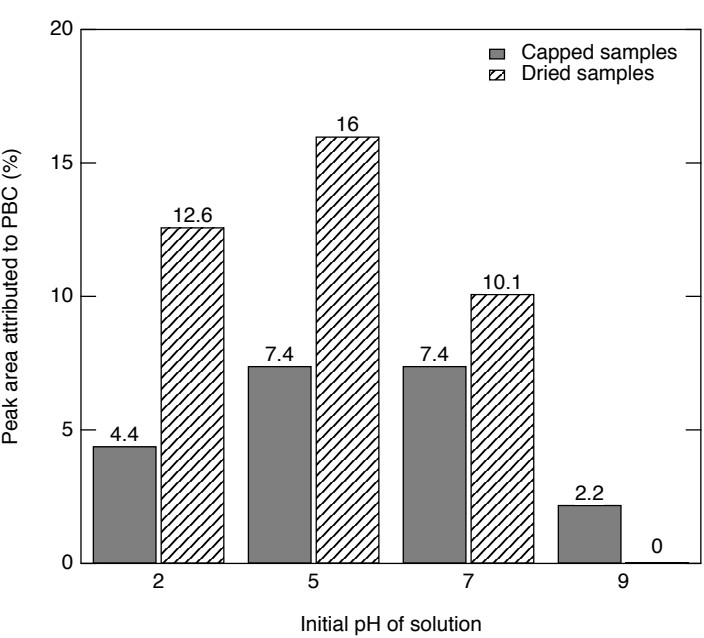

**Figure 6.** Estimated contribution of PBC to the total ion signal calculated as the sum of the "percent peak areas" for masses m/z 109, 162, 181, 235, 253, and 289. Many of these masses were below detection limit in one or more sample.





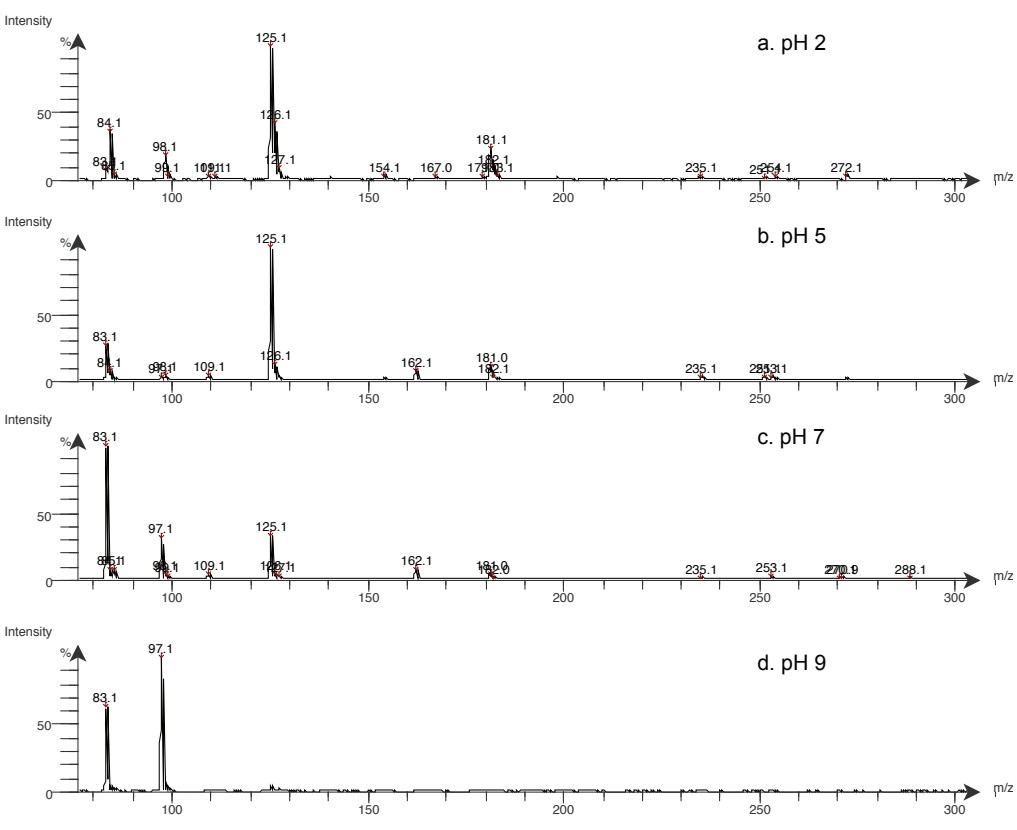

**Figure 7.** Atmospheric pressure chemical ionization (APCI) mass spectra for all four initial pH conditions in dried samples. Intensity is set to 100% for the largest signal and does not correspond to concentration in the sample itself.





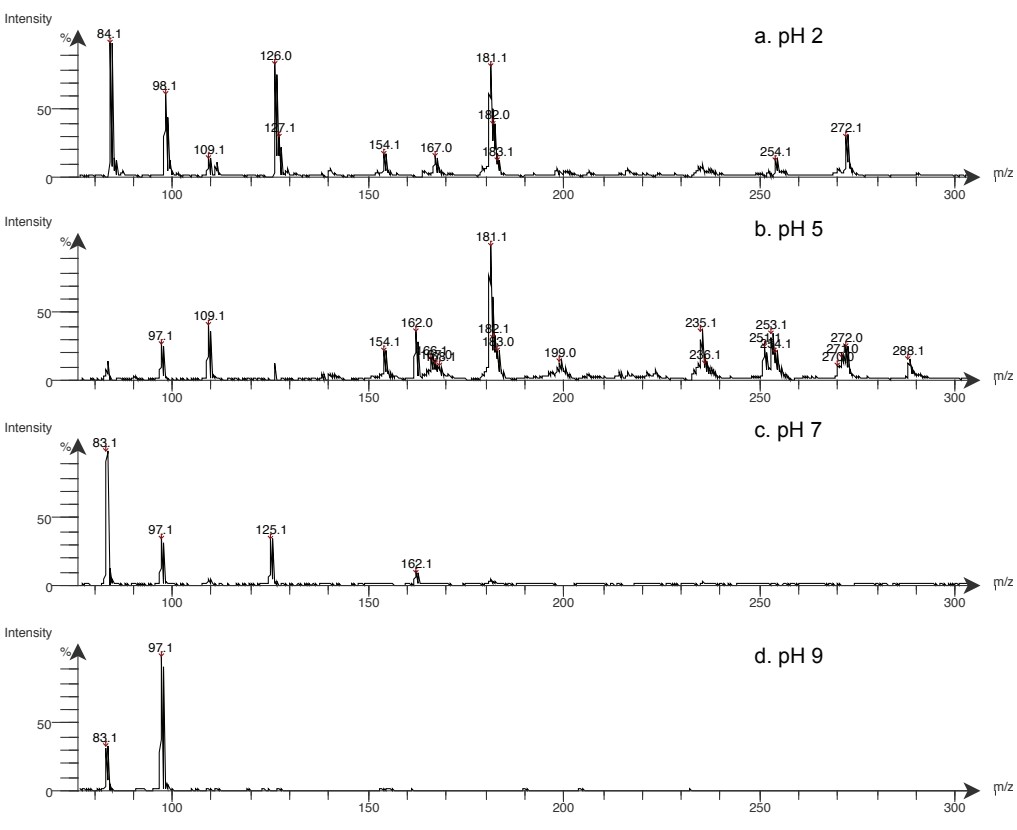

**Figure 8.** Atmospheric pressure chemical ionization (APCI) mass spectra for all four initial pH conditions in dried samples after subtraction of capped sample spectra for each corresponding sample. Intensity is set to 100% for the largest signal after subtraction and does not correspond to concentration in the sample itself.





**Scheme 1** Proposed formation mechanism of 2,5-dimethylpyrazine (**1**) from methylglyoxal and ammonium sulfate under mildly acidic conditions and in the presence of a reducing agent (formic acid) and atmospheric oxygen. This mechanism is slightly adapted from Figure 4 of Divine et al. (2012) which begins with the accepted formation of 2,5-dimethylpyrazine described in Rizz (1972). A plausible alternative mechanism is the reduction of only one ketoimine and a subsequent heteroisomerization, eliminating the need for the final oxidation step.





**Scheme 2** Proposed formation mechanism of pyrazine-based chromophores **2**, **3**, and **4** with m/z 181, 253, and 235 observed by APCI from 2,5-dimethylpyrazine and methylglyoxal. Addition of methylglyoxal here follows the mechanism presented in De Haan et al. (2011) for oligomerization of imidizole-based products in this same system. Red and blue are used to highlight individual and intact methylglyoxal units in the products that were directly observed by APCI-MS. Additional possible structures for m/z 253 and m/z 235 are shown in Table 3.


**Scheme 3** Proposed formation mechanism of **5** with m/z 162 from **2** (m/z 181) involving imine substitution and dehydration.

**Table 1.** Absorptivity (as mass absorption coefficient, MAC) and absorption Ångström exponent (AAE) for dried and capped samples for each initial pH condition.

| Sample | $\text{MAC}_{365}$ ($\text{m}^2\ \text{g}^{-1}$) | AAE |
|---|---|---|
| Capped samples | | |
| pH 2 | 0.05 | 9 |
| pH 5 | 0.12 | 11 |
| pH 7 | 0.17 | 12 |
| pH 9 | 0.22 | 12 |
| Dried samples | | |
| pH 2 | 0.59 | 7.7 |
| pH 5 | 0.75 | 8.6 |
| pH 7 | 0.78 | 8.9 |
| pH 9 | 1.13 | 8.0 |





**Table 2.** Proposed structures for products detected by APCI-MS that are either (a) previously reported or (b) analogs of previously reported compounds. In the case of type (a) compounds, citations are provided. The observed m/z in $^{15}$N samples is provided in parentheses.

| m/z observed (in $^{15}$N) | Molecular formula of ion | Proposed structure of ion(s) | Previous observations and prevalence in this study |
|---|---|---|---|
| 83 (85) | $C_4H_7N_2^+$ | | [a]De Haan et al., 2011; 2nd largest signal in most samples, highest in pH 9. |
| 97 (99) | $C_5H_9N_2^+$ | | [a]De Haan et al., 2011; Only observed above pH 7. |
| 125 (127) | $C_6H_9N_2O^+$ | | [a]De Haan et al., 2011; Lin et al., 2015; Largest signal in all samples except pH 9. |
| 144 (145) | $C_6H_{10}NO_3^+$ | | [a]Lin et al., 2015; Minor to v. minor, stronger signal in capped vials. |
| 180 (181) | $C_9H_{10}NO_3^+$ | | [b]Double dehydration product of m/z 216; V. minor, mostly observed in dried samples. |
| 197 (199) | $C_9H_{13}N_2O_3^+$ | | [a]Lin et al., 2015; Minor, stronger signal in capped vials. |
| 216 (217) | $C_9H_{14}NO_5^+$ | | [b]MG add'n product of m/z 144; V. minor, only observed in pH 7 and 9 capped samples. |
| 232 (235) | $C_{12}H_{14}N_3O_2^+$ | | [a]Lin et al., 2015, Aiona et al., 2017; Very minor, found in dried vials. |
| 234 (235) | $C_{12}H_{12}NO_4^+$ | | [b]Triple dehydration from m/z 288; Appears in pH 7-9 samples. |
| 251 (253) | $C_{12}H_{15}N_2O_4^+$ | | [a]Lin et al., 2015; Consistently observed in dried samples. |
| 288 (289) | $C_{12}H_{18}NO_7^+$ | | [b]Two MG add'n product of m/z 144; Minor, favored in capped samples. |





**Table 3.** Proposed structures for novel pyrazine-based chromophores matching the molecular weight and number of N atoms observed in this study. The observed m/z in $^{15}$N samples is provided in parentheses. In some cases, the ion (but not the structure) was reported in a previous study.

| m/z observed (in $^{15}$N) | Molecular formula of ion | Proposed structure of ion(s) | Previous observations and prevalence in this study |
|---|---|---|---|
| 109 (111) | $C_6H_9N_2^+$ | | Peak 9 in Lin et al., 2015; Minor but consistently observed in dried, acidic samples. |
| 162 (165) | $C_9H_{12}N_3^+$ | | Peak 18 in Lin 2015; Minor but consistently observed in most samples. 4th largest signal in pH 7 dried sample. |
| 181 (183) | $C_9H_{13}N_2O_2^+$ | | Minor but consistently observed in most samples. 10% of total signal in pH 2 dried sample. |
| 235 (237) | $C_{12}H_{15}N_2O_3^+$ | | Peak 1 in Lin et al. 2015; Minor but consistent, significantly larger in dried samples. Regioisomer not shown. |
| 253 (255) | $C_{12}H_{17}N_2O_4^+$ | | Peak 14 in Lin et al. 2015; Minor, slightly favored in capped samples over dried. |
| 289 (291) | $C_{15}H_{17}N_2O_4^+$ | | Very minor, observed in dried samples only. Regioisomer not shown. |