# Peer review of "Evidence for pyrazine-based chromophores in cloudwater mimics containing methylglyoxal and ammonium sulfate"

_Atmospheric Chemistry and Physics, 2018_

## Referee Comment (RC1) · Anonymous Referee #1 · 17 Apr 2018

The authors present the results of an experimental study of brown carbon formation in solutions intended to mimic cloud water which contained methylglyoxal and ammonium sulfate. BrC formation was studied at a range of pH, and for some solutions which were allowed to evaporate into a hood over the course of 7 days.

The authors use chemical ionization mass spectrometry to analyze products formed, and while the mass/charge ratios of many of the identified compounds had been identified earlier in other laboratory studies of this system in the literature, by performing experiments with isotopically labeled ammonium sulfate, the authors were able to distinguish the number of nitrogens in each species. This analysis led them to propose

a new type of light-absorbing organic compound which has not been previously identified in atmospheric aerosols or their laboratory proxies, pyrazines. This element of the study is a valuable new contribution.

I have a number of serious concerns about the evaporation experiments which I am not sure can be addressed. It may be necessary to cut this material from the paper, or heavily revise with additional control experiments. It seems that the experimental method involved leaving samples uncovered in a hood for a week.

- Cloud processing takes place on a timescale of minutes to hours, so this process does not resemble the experimental conditions. How does the timescale for drying affect the results? How would one derive quantitative kinetic information from this complex combined reaction/dehydration process, and how could it be justified as being similar to what actually happens in the atmosphere?

- Leaving samples uncovered in the lab is known to lead to BrC formation in SOA samples due to contamination (e.g. the early preliminary data of Bones et al. JGR 2010). How can the authors eliminate the possibility that contamination contributed to the enhanced absorption in the dried and reconstituted samples?

- It seems unnecessary to specifically compare the effects of pH and evaporation (line 34 page 3) when mechanistically these processes are distinct and not in competition with each other in ambient cloud droplets. It's relevant to quantify both processes, but I doubt a meaningful direct comparison can be made based on the data here.

Specific comments

Abstract, line 3 - why is Maillard type in parentheses
* * *

---

## Author Comment (AC1) · 22 Apr 2018

Author Response to RC1

We thank the first reviewer for providing these helpful comments. Although we are in the process of completing the work to address the reviewer's reasonable concerns, we wanted to post a response immediately. The additional experiments are not overly burdensome and should address the concerns completely.

RC1: "I have a number of serious concerns about the evaporation experiments which I am not sure can be addressed. It may be necessary to cut this material from the paper,

or heavily revise with additional control experiments. It seems that the experimental method involved leaving samples uncovered in a hood for a week."

Yes, the reviewer is correct that for the evaporated samples, the vials are left in a clean, unused hood in laboratory used only by the PI for one week. We can address a number of the reviewer's concerns with measurements we already have (that were not included in the original submission) and are happy to conduct additional control experiments for the final publication to confirm that the evaporation process does not significantly differ from droplet scale evaporation experiments. We believe that the findings regarding evaporation are important enough to merit further attention, and so hope they can be left in with the proposed additions.

Specific responses to individual concerns are addressed below.

1. "Cloud processing takes place on a timescale of minutes to hours, so this process does not resemble the experimental conditions. How does the timescale for drying affect the results?"

As the experiments were conducted, we analyzed the sample material after just a few hours, 24 hours, and 72 hours. In the case of the capped samples, the only change we observed from 24 hours to 1 week was the increase of all signals. All the products we observed after 1 week were visible in the initial spectra (will be added to the SI). The convenience of the 1 week old samples is purely in the signal to noise ratio of the spectra and easier interpretation of isotopically labelled products. As for the dried samples (more of a concern, I suspect), we see something different. The samples appear to change the product distribution over the final hours of drying resulting in the noticeable difference between capped and dried samples. Until the open samples actually dry, they resemble the capped samples (again, spectra will be added to the SI). The actual transition from a solution to dried material happens fairly quickly, despite the samples being allowed to dry for a week. Once dried, the samples do not change composition measurably, and we have verified this by analyzing dried material immediately following drying, and a week later. To compare this week long drying to rapid drying, we will conduct some additional experiments under two different mechanisms. First, we will dry samples (prepared in the same way) under high purity nitrogen. Past experience doing this suggests that drying will take place within 3 hours. Second, we will atomize a solution diluted by a factor of 1/100, dry the droplets by diffusion, and collect the particles by impaction onto the glass capillary used for APCI so that the dried particles will be directly analyzed following atomization. Again, preliminary work not included in the original submission suggests that the products will be the same but that the signal will be low. We will be sure to collect enough material to verify that the drying process in the hood has only served to increase the sample size/signal and not to distort the products. Any differences will be described in the manuscript text.

Using long drying time (or rotary evaporation with low heat) to understand the effect of cloud processing is not unique to this work (Nguyen et al. 2012; Powelson et al., 2013; Aiona et al., 2017), though the concerns about its relevance are the reason that cloud chamber facilities are desirable (when available) as the role of surface chemistry is not entirely known and sure to affect the product distribution somewhat. We intend to conduct further studies using atomization onto APCI capillary probes, with internal standards, to look further into this issue in our next study.

2. "How would one derive quantitative kinetic information from this complex combined reaction/dehydration process, and how could it be justified as being similar to what actually happens in the atmosphere? "

In order to obtain kinetic information, the study would have to performed with a more quantitative method for determining product concentrations which necessarily requires standards of these compounds to that ionization efficiency can be determined. Only 2,5-DMP is available as a standard – the other products would have to be purified and a response in APCI quantified. While deriving kinetics is beyond the scope of this work, we hope that future studies will target one or more of the pyrazine products here for quantitative kinetic analyses. It is possible to use GC-MS, but that requires

extraction of these products into more volatile solvents (much like the food studies included in our references). An alternative might method might involve the use of an internal standard, such as pyrazine, that was not observed in our samples but might have similar ionization efficiency to 2,5-DMP. We propose that one might easily do a kinetic study on the capped samples, but that the kinetics of evaporation are far more difficult to quantify.

3. "Leaving samples uncovered in the lab is known to lead to BrC formation in SOA samples due to contamination (e.g. the early preliminary data of Bones et al. JGR 2010). How can the authors eliminate the possibility that contamination contributed to the enhanced absorption in the dried and reconstituted samples?" This is a good point – we did conduct control experiments with only methylglyoxal that was not included in the original submission. We will do a similar study with AS. We will include those in the revised SI to illustrate the difference in browning between the mixtures and the control experiments. Indeed, we do see a small amount of contamination, but it represents only a fraction of the browning in the mixtures. We are adding this to the text of the manuscript as well.

4. "It seems unnecessary to specifically compare the effects of pH and evaporation (line 34 page 3) when mechanistically these processes are distinct and not in competition with each other in ambient cloud droplets. It's relevant to quantify both processes, but I doubt a meaningful direct comparison can be made based on the data here."

We agree that the pH and evaporation processes are not in competition with one another; our point was more that the evaporation process can produce material with the absorptivity (or substantially greater absorptivity) than the effect of high pH. In previous studies (Yu et al., 2011; Kampf et al., 2012), the authors have asserted that this Maillard-type chemistry has limited brown carbon potential due to the acidic nature of atmospheric water and the unfavorable rate of these reactions under acidic conditions compared to basic conditions. While true, we assert that the effect of evaporation in forming brown carbon chromophores is so strong as to produce material with more

absorptivity in pH 2 dried samples than that observed at pH 9 (arguably the most favorable for nucleophilic attack by ammonia). When the role of evaporation is correctly accounted for, we proposed that these reactions can in fact contribute to atmospheric brown carbon in acidic cloud and aerosol water. We will edit the manuscript text to make this point clearer.

Specific comment regarding the use of "Maillard type" – we used parentheses to distinguish the use of ammonium sulfate from intact amino acids, as is standard for Maillard reactions in food studies. But, we have omitted the parentheses in the revised version.

References cited in this response:

Aiona, P. K., Lee, H. J., Lin, P., Heller, F., Laskin, A., Laskin, J., and Nizkorodov, S. A.: A Role for 2-Methyl Pyrrole in the Browning of 4-Oxopentanal and Limonene Secondary Organic Aerosol, Environmental science & technology, 51, 11 048–11 056, 2017b.

Kampf, C. J., Jakob, R., and Hoffmann, T.: Identification and characterization of aging products in the glyoxal/ammonium sulfate system– implications for light-absorbing material in atmospheric aerosols, Atmospheric Chemistry and Physics, 12, 6323–6333, 2012.

Nguyen, T. B., Lee, P. B., Updyke, K. M., Bones, D. L., Laskin, J., Laskin, A., and Nizkorodov, S. A.: Formation of nitrogen-and sulfur-containing light-absorbing compounds accelerated by evaporation of water from secondary organic aerosols, Journal of Geophysical Research: Atmospheres (1984–2012), 117, 2012.

Powelson, M. H., Espelien, B. M., Hawkins, L. N., Galloway, M. M., and De Haan, D. O.: Brown carbon formation by aqueous-phase carbonyl compound reactions with amines and ammonium sulfate, Environmental science & technology, 48, 985–993, 2013.

Yu, G., Bayer, A. R., Galloway, M. M., Korshavn, K. J., Fry, C. G., and Keutsch, F. N.: Glyoxal in aqueous ammonium sulfate solutions: products, kinetics and hydration effects, Environmental science & technology, 45, 6336–6342, 2011.

---

## Referee Comment (RC2) · Anonymous Referee #4 · 13 May 2018

Hawkins et al. present a laboratory experiment to simulate the formation of aqueous-phase brown carbon (aqBrC) from methylglyoxal and ammonium sulfate. The reactions of methylglyoxal (or glyoxal) + ammonium sulfate (or amines and amino acids) have been employed as a canonical chemical system to mimic BrC formation. Despite numerous studies on this topic, the chemical insights of the chromophores remain unclear. Using an innovative APCI technique with isotopically labeled ammonium sulfate, the authors present convincing evidence for pyrazine-based chromophores present in the reaction mixture. The chemical analyses are highly detailed. I believe that the molecular-level information on BrC chromophores presented in this work will lay the foundation for understanding the environmental impact of BrC. I strongly recommend publication of this work in Atmospheric Chemistry and Physics. However, I have several comments/suggestions to improve the manuscript.

Major comments

- I am concerned that the conclusion "droplet evaporation can overcome the pH barrier" is an overstatement. Personally, I think there is no doubt that evaporation accelerates BrC formation. For example, Lee et al.[1] have shown that diffusion-drying atomized droplets gave rise to BrC within seconds (I think the authors should consider citing this paper). However, the current experimental approach (i.e., free-drying in a vial for 7 days) cannot conclusively show that droplet drying can overcome acidity barrier under atmospherically realistic conditions (i.e., rapid drying, evaporation of volatile compounds).
- The authors only present APCI spectra after one week of drying. Something missing from the current analytical protocol is the initial MS and TOC concentration (i.e., immediately after mixing of methylglyoxal and AS). It may seem trivial, but it is essential to show that the peaks presented by the authors are indeed from the reaction. It is also a good way to test whether APCI is indeed insensitive to methylglyoxal.
- Despite a detailed discussion on the effects of evaporation, the authors seem to have neglected the fact that evaporation also occurs during the APCI measurement (from the ASAP probe). In particular, the capped samples are in liquid while the dried samples are in solid. The authors should discuss how this may affect the APCI interpretation.

Minor and technical comments

- Page 1, line 4: including in their complexity
- Page 2, line 22: Fu 2008 is not properly cited and does not show in the bibliography.
- Page 4, line 5: "TOC" appears for the first time here but is not properly introduced. Meanwhile, it is introduced later twice (Page 4, lines 16 and 19).
- Page 5, line 21: "with clear shoulders at 350". It appears to me that the capped samples also exhibit shoulders at 350 nm. It is more obvious from the log-log figure (Figure 1B). If the shoulder is not an important part of the discussion, I would suggest that the authors removed this part.
- Page 6 Line 12 (awkward sentence): "The masses observed in the pH 2 samples in Figure 2 compose most but not all of the products observed across all samples though the ratio of products is pH-dependent."
- Page 6 line 29: 2,5-dimethylpyrazine appears here for the first time, hence (2,5-DMP) should be defined here instead of line 30.
- Page 6, line 31: "m/z 162 is proposed as an imine-substituted dehydration product of m/z 181 formed as shown in Scheme 3." The authors refer to Scheme 3 here before Scheme 1 or 2. I would suggest a reorder of either the sentences or the schemes.

- Page 10, line 28: "Because the absorptivity metric accounts for OC concentration, we cannot simply attribute the higher absorptivity to higher a concentration of chromophores driven by the faster reaction rate under basic conditions." This statement is puzzling. The TOC concentrations in the capped samples should remain relatively constant regardless of the initial pH. Even after normalization with TOC (equation 1), the higher absorptivity may still simply represent the faster formation of BrC in basic conditions. I think the current statement is correct only if the absorption were to be normalized against BrC concentration (which cannot be quantified).
- Although the evidence for PBC is strong, it seems that PBCs do not contribute to the strong light absorptivity at under more basic conditions. In Conclusion, the authors should briefly comment on the current "unknowns" for aqBrC. This should serve as a guidance for future research.

Reference

(1) Lee, A. K.; Zhao, R.; Li, R.; Liggio, J.; Li, S.-M.; Abbatt, J. P. Formation of Light Absorbing Organo-Nitrogen Species from Evaporation of Droplets Containing Glyoxal and Ammonium Sulfate. *Environ. Sci. Technol.* **2013**, *47* (22), 12819–12826.

---

## Author Comment (AC2) · 15 May 2018

Author Response to RC2

RC2: "Hawkins et al. present a laboratory experiment to simulate the formation of aqueous-phase brown carbon (aqBrC) from methylglyoxal and ammonium sulfate. The reactions of methylglyoxal (or glyoxal) + ammonium sulfate (or amines and amino acids) have been employed as a canonical chemical system to mimic BrC formation. Despite numerous studies on this topic, the chemical insights of the chromophores remain unclear. Using an innovative APCI technique with isotopically labeled ammonium sulfate, the authors present convincing evidence for pyrazine-based chromophores

present in the reaction mixture. The chemical analyses are highly detailed. I believe that the molecular-level information on BrC chromophores presented in this work will lay the foundation for understanding the environmental impact of BrC. I strongly recommend publication of this work in Atmospheric Chemistry and Physics. However, I have several comments/suggestions to improve the manuscript."

The authors are grateful for the thoughtful comments and find the major suggestions to be entirely reasonable and manageable in the time frame provided. Indeed, some of the suggested measurements have been completed already, but were carelessly excluded from the SI document upon submission.

Specific responses to individual concerns are addressed below.

1. "I am concerned that the conclusion "droplet evaporation can overcome the pH barrier" is an overstatement. Personally, I think there is no doubt that evaporation accelerates BrC formation. For example, Lee et al.1 have shown that diffusion-drying atomized droplets gave rise to BrC within seconds (I think the authors should consider citing this paper). However, the current experimental approach (i.e., free-drying in a vial for 7 days) cannot conclusively show that droplet drying can overcome acidity barrier under atmospherically realistic conditions (i.e., rapid drying, evaporation of volatile compounds)."

The authors agree with this assessment and have planned to both modify the text to perform preliminary studies using an atomizer to determine if any of the observed product are visible upon rapid drying. These measurements will be included in the SI.

2. "The authors only present APCI spectra after one week of drying. Something missing from the current analytical protocol is the initial MS and TOC concentration (i.e., immediately after mixing of methylglyoxal and AS). It may seem trivial, but it is essential to show that the peaks presented by the authors are indeed from the reaction. It is also a good way to test whether APCI is indeed insensitive to methylglyoxal."

The authors did obtain spectra immediately after mixing as well as 24 and 72 hours later, those will be included in the SI along with the blanks suggested by Reviewer #1. Initial TOC measurements as well as absorbance spectra will be collected on new samples prepared prior to submission of the revised draft.

3. "Despite a detailed discussion on the effects of evaporation, the authors seem to have neglected the fact that evaporation also occurs during the APCI measurement (from the ASAP probe). In particular, the capped samples are in liquid while the dried samples are in solid. The authors should discuss how this may affect the APCI interpretation."

The authors agree that evaporation during APCI analysis could be important in product formation. A discussion of this will be added to the revised manuscript. While nothing can be done to prevent the liquid samples from evaporating during analysis, we were able to re-dissolve the dried material and collect spectra to compare with the dried (original) material. Those spectra were identical within instrumental uncertainty. In almost all cases, the capillary became coated with a dark brown material after exposure to heat and dry N2, indicating that the method does indeed generate additional brown products beyond reaction. However, the observance of masses matching our pyrazine chromophores in previous studies using other methods (and in our GC-MS samples that never experience drying) gives us confidence in the formation of pyrazine chromophores prior to APCI analysis.

The authors are willing to make all of the suggested minor and technical corrections.

---

## Author Response (AR1)

A note about the LatexDiff PDF file
I used the program LatexDiff to produce the annotated revision file for the manuscript. It seems to have worked. However, in one location the difference program indicates that I deleted a string of references that I did not delete (page 10 of the combined PDF for reviewers, line 19, beginning with Nguyen et al., 2012). This can be verified by simply looking at the new (unmarked) manuscript file. The references are still there.

Regarding the SI file, I simply reordered the figures and added several more so I did not run the LatexDiff code on the SI. I did include the SI in the combined package for reviewers because many of their requests resulted in new figures in the SI.

Author Response to RC1

*RC1: "I have a number of serious concerns about the evaporation experiments which I am not sure can be addressed. It may be necessary to cut this material from the paper, or heavily revise with additional control experiments. It seems that the experimental method involved leaving samples uncovered in a hood for a week."*

Yes, the reviewer is correct that for the evaporated samples, the vials are left in a clean, unused hood in laboratory used only by the PI for one week. We can address a number of the reviewer's concerns with measurements now added to the SI and the remaining concerns are addressed in the revised text.

Specific responses to individual concerns are addressed below.

1. *"Cloud processing takes place on a timescale of minutes to hours, so this process does not resemble the experimental conditions. How does the timescale for drying affect the results?"*

   As the experiments were conducted, we analyzed the sample material after just a few hours, 24 hours, and 72 hours. In the case of the capped samples, the only *chemical* change we observed from 24 hours to 1 week was the increase of all signals. Many of the products we observed after 1 week were visible in the initial spectra, though the signal was very small (Fig 3 and Figures S4-S7). In contrast, the formation of imidazole (m/z 125) is visible immediately. Further, after only 10 minutes we observe the disappearance of peaks related to methylglyoxal (m/z 43, 57, 75 in Fig 3). The use of 1 week old samples improves signal/noise ratios for minor products and allows easier interpretation of isotopically labelled products. We have added text regarding the impact of reaction time on the atmospheric relevance of these products. However, we do have evidence that while some of the products are not forming on the order of minutes, many are forming in less than 1 day (described further below).

As for the dried samples, we see something different. The product distribution of the samples appears to change over the final hours of drying resulting in the noticeable difference between capped and dried samples. The issue of contamination is addressed in question 3; here we comment on the timescale only.

To compare the week-long drying to rapid drying on atmospherically relevant timescales, we conducted some additional experiments by two new methods. First, we dried samples (prepared in the same way but only at pH 5) either under high purity nitrogen or HEPA-filtered lab air. The process was completed within 1 hour. We analyzed the chemical composition as well as UV/visible absorptivity immediately and 24 hours later. UV/visible spectra are provided in Figure S13 while APCI spectra are included in Figures S14-S15. We observed that indeed, no sample became as absorptive as the sample dried over 7 days (Fig S13). However, within 24 hours of rapid drying, both the air and nitrogen dried samples became significantly more absorptive beyond 350 nm than the sample capped for one week. In our next study, we intend to quantify this rate of browning to determine the possibility of these chromophores forming on timescales of hours as the reviewer suggests.

In the APCI spectra of these fast-dried samples shown in Figures S14-S15, we see the methylimidazole product (m/z 125) as well as m/z 162, m/z 181, and m/z 235 (all proposed pyrazine-based chromophores). In fact, comparing Figure S14 to panel c in Figure S3, it is clear that the week-long drying creates the same products as the rapid drying. Given the difference in absorptivity we observed, it is likely that the relative concentration of strong chromophores is changing from 24 hrs to 7 days.

In a second experiment, we atomized a solution diluted by a factor of 1/20 compared to the original evaporation experiments, dried the droplets by diffusion, and collected the particles by impaction onto the glass capillary used for APCI so that the dried particles were directly analyzed following atomization (Fig S16). Interestingly, m/z 125 was not the most abundant ion observed. Instead, m/z 234 ($C_{12}H_{12}NO_4^+$, a linear imine based compound) is the most abundant. Related compounds, m/z 144 and 306, are also observed. In addition, methylglyoxal self-reaction products are visible (m/z 199).

Using time (or rotary evaporation) to understand the effect of cloud processing is not unique to this work (Nguyen et al. 2012; Powelson et al., 2013; Aiona et al., 2017), though the concerns about its relevance are the reason that cloud chamber facilities are desirable (when available) as the role of surface chemistry is not entirely known and sure to affect the product distribution somewhat. We intend to conduct further studies using atomization onto APCI capillary probes, with internal standards, to look further into this issue.

2. *"How would one derive quantitative kinetic information from this complex combined reaction/dehydration process, and how could it be justified as being similar to what actually happens in the atmosphere?"*

In order to obtain kinetic information by APCI, the study would have to performed with a more quantitative method for determining product concentrations which necessarily requires standards of these compounds so that ionization efficiency can be determined. Only 2,5-DMP is available as a standard – the other products would have to be purified and a response in APCI quantified. However, it is possible to use NMR to determine product formation assuming that the shift of pyrazine protons is significantly removed from the imidazole and other products. While deriving kinetics is beyond the scope of this work, we hope that future studies will target one or more of the pyrazine products here for quantitative kinetic analyses. It is also possible to use GC-MS, but that requires extraction of these products into more volatile solvents (much like the food studies included in our references). An alternative method might involve the use of an internal standard, such as pyrazine, that was not observed in our samples but might have similar ionization efficiency to 2,5-DMP. We propose that one might easily do a kinetic study on the capped samples, but that the kinetics of evaporation are far more difficult to quantify. Certainly, this is something worth pursuing.

3. *"Leaving samples uncovered in the lab is known to lead to BrC formation in SOA samples due to contamination (e.g. the early preliminary data of Bones et al. JGR 2010). How can the authors eliminate the possibility that contamination contributed to the enhanced absorption in the dried and reconstituted samples?"*

This is a good point – we did conduct control experiments with a) only methylglyoxal and b) only ammonium sulfate that were not included in the original submission. We repeated those studies and have included those now in the revised SI (Figures 1b, S8, and S9) to illustrate the difference in composition and browning between the mixtures and the control experiments. Ammonium sulfate did not show any signal in either the TOC analysis (OC levels were similar to blanks) or the UV/visible spectra. However, we did observe some chromophore formation with only methylglyoxal, particularly at pH 9 (Fig 1b). The absorptivity is greatest for the pH 9 methylglyoxal sample regardless of whether the sample was uncovered or covered. For the pH 2-7 samples, very little absorbance was observed, amounting to smaller mass absorption coefficients than the pH 2 capped sample after 1 week. It is worth noting that the pH 9 samples are the least atmospherically relevant, though including those experiments helps elucidate the role of pH in these reactions.

Looking at the APCI mass spectra for the methylglyoxal control samples, we see

evidence of acid-catalyzed reactions generating larger products (S8a and S8b) while the neutral and basic samples favor smaller products. Therefore, it seems likely that the intense color observed at pH 9 for the methylglyoxal samples arises from a few deeply colored chromophores or from products that are efficiently volatilized or ionized by APCI.

4. *"It seems unnecessary to specifically compare the effects of pH and evaporation (line 34 page 3) when mechanistically these processes are distinct and not in competition with each other in ambient cloud droplets. It's relevant to quantify both processes, but I doubt a meaningful direct comparison can be made based on the data here."*

We agree that the pH and evaporation processes are not in competition with one another; our point was more that the evaporation process can produce material with the same absorptivity (or substantially greater absorptivity) than the effect of high pH. As we addressed above, this assertion has an obvious caveat (time of reaction). In previous studies (Yu et al., 2011; Kampf et al., 2012), the authors have asserted that these Maillard reactions have limited potential to form atmospheric brown carbon due to the acidic nature of atmospheric water and the unfavorable rate of these reactions under acidic conditions compared to basic conditions. While true, we assert that the effect of evaporation in forming brown carbon chromophores is so strong as to produce material with more absorptivity in pH 2 dried samples than that observed at pH 9 (arguably the most favorable for nucleophilic attack by ammonia). When the role of evaporation is correctly accounted for, we proposed that these reactions can in fact contribute to atmospheric brown carbon in acidic cloud and aerosol water.

Given the lower (but still significant) absorptivity observed in our fast-dried samples, the extent to which speed of evaporation limits brown carbon formation must be addressed. The manuscript has been edited to reflect this limitation and to suggest this question for future work.

Specific comment regarding the use of "Maillard type" – we used parentheses to distinguish the use of ammonium sulfate from intact amino acids, as is standard for Maillard reactions in food studies. But, we have omitted the parentheses in the revised version.

References cited in this response:
Aiona, P. K., Lee, H. J., Lin, P., Heller, F., Laskin, A., Laskin, J., and Nizkorodov, S. A.: A Role for 2-Methyl Pyrrole in the Browning of 4-Oxopentanal and Limonene Secondary Organic Aerosol, Environmental science & technology, 51, 11 048–11 056, 2017b.

Kampf, C. J., Jakob, R., and Hoffmann, T.: Identification and characterization of aging products in the glyoxal/ammonium sulfate system– implications for light-absorbing material in atmospheric aerosols, Atmospheric Chemistry and Physics, 12, 6323–6333, 2012.

Nguyen, T. B., Lee, P. B., Updyke, K. M., Bones, D. L., Laskin, J., Laskin, A., and Nizkorodov, S. A.: Formation of nitrogen-and sulfur-containing light-absorbing compounds accelerated by evaporation of water from secondary organic aerosols, Journal of Geophysical Research: Atmospheres (1984–2012), 117, 2012.

Powelson, M. H., Espelien, B. M., Hawkins, L. N., Galloway, M. M., and De Haan, D. O.: Brown carbon formation by aqueous-phase carbonyl compound reactions with amines and ammonium sulfate, Environmental science & technology, 48, 985–993, 2013.

Yu, G., Bayer, A. R., Galloway, M. M., Korshavn, K. J., Fry, C. G., and Keutsch, F. N.: Glyoxal in aqueous ammonium sulfate solutions: products, kinetics and hydration effects, Environmental science & technology, 45, 6336–6342, 2011.

Author Response to RC2

*RC2: "Hawkins et al. present a laboratory experiment to simulate the formation of aqueous-phase brown carbon (aqBrC) from methylglyoxal and ammonium sulfate. The reactions of methylglyoxal (or glyoxal) + ammonium sulfate (or amines and amino acids) have been employed as a canonical chemical system to mimic BrC formation. Despite numerous studies on this topic, the chemical insights of the chromophores remain unclear. Using an innovative APCI technique with isotopically labeled ammonium sulfate, the authors present convincing evidence for pyrazine-based chromophores present in the reaction mixture. The chemical analyses are highly detailed. I believe that the molecular-level information on BrC chromophores presented in this work will lay the foundation for understanding the environmental impact of BrC. I strongly recommend publication of this work in Atmospheric Chemistry and Physics. However, I have several comments/suggestions to improve the manuscript."*

The authors are grateful for the thoughtful comments and find the major suggestions to be entirely reasonable and manageable in the time frame provided. The necessary information has been added to the SI and in some cases, to the manuscript figures themselves.

Specific responses to individual concerns are addressed below.

1. *"I am concerned that the conclusion "droplet evaporation can overcome the pH barrier" is an overstatement. Personally, I think there is no doubt that evaporation accelerates BrC formation. For example, Lee et al.[1] have shown that diffusion-drying atomized droplets gave rise to BrC within seconds (I think the authors should consider citing this paper). However, the current experimental approach (i.e., free-drying in a vial for 7 days) cannot conclusively show that droplet drying can overcome acidity barrier under atmospherically realistic conditions (i.e., rapid drying, evaporation of volatile compounds)."*

The authors agree with this assessment and have modified the text to address this limitation as well as performed preliminary studies using an atomizer to determine if any of the observed products are visible upon rapid drying. These measurements are now included in the SI. Descriptions of the new measurements and the implications for this work are described in the response to reviewer #1 (who had the same concern). We have added the paper from Lee et al., in 2013 to our discussion and introduction as it nicely demonstrates a number of the points made here.

2. *"The authors only present APCI spectra after one week of drying. Something missing from the current analytical protocol is the initial MS and TOC concentration (i.e., immediately after mixing of methylglyoxal and AS). It may seem trivial, but it is essential to show that the peaks presented by the authors are indeed from the reaction. It is also a good way to test whether APCI is indeed insensitive to methylglyoxal."*

Yes, this was an oversight in the original submission and we thank the reviewer for pointing this out. We have now included both UV/visible absorptivity analysis and mass spectra for the reactions after 10 minutes and for methylglyoxal alone in Figure 1 of the manuscript. Additional control experiments and initial measurements are in the supporting information. Absorbance spectra are included in Figure 1 while initial APCI spectra are included in S4-7. Samples containing only methylglyoxal are included in S8-S9. Figure 1a illustrates that the absorptivity reported for the reactions mixtures is indeed due to the reaction. Figures S4 and S7 illustrate that the mixtures are immediately different from methylglyoxal alone (at pH 2 and 9, respectively) and that methylglyoxal is indeed observable at m/z 75 (as hydroxyacetone, the product of reduction of methylglyoxal). In addition, two fragments are visible (m/z 57 and m/z 43, loss of water and CHO respectively). We have edited the text of the manuscript to reflect the observation of methylglyoxal in the controls.

3. *"Despite a detailed discussion on the effects of evaporation, the authors seem to have neglected the fact that evaporation also occurs during the APCI measurement (from the ASAP probe). In particular, the capped samples are in liquid while the dried samples are in solid. The authors should discuss how this may affect the APCI interpretation."*

The authors agree that evaporation during APCI analysis could be important in product formation. A discussion of this has been added to the revised manuscript. This is especially relevant given the appearance of pyrazines during the cooking process. And in fact, in many cases, the capillary became coated with a dark brown material after exposure to heat and dry $N_2$, indicating that the method does indeed generate additional brown products beyond reaction.

However, the observance of masses matching our pyrazine chromophores in previous studies using other methods (and in our GC-MS samples that never experienced heating) gives us confidence in the formation of pyrazine chromophores prior to APCI analysis. We have planned future experiments to carefully quantify pyrazine and 2,5-dimethylpyrazine in these and similar reactions by GC-MS with the aim of addressing the concern regarding enhanced brown carbon formation during APCI analysis.

The authors have made all of the suggested minor and technical corrections.

[revised manuscript text omitted]

a. Background after EtOH cleaning

[Figure]

b. 1.0 M methylglyoxal + 1.0 M ammonium sulfate, 10 minutes after mixing, approx pH 5

**Figure S5.** Methylglyoxal (1.0 M) and ammonium sulfate (1.0 M) reactions, adjusted to pH 5 analyzed by APCI after 10 min using low temperature and low fragmentation conditions.

[Figure]

a. Background after EtOH cleaning

[Figure]

b. 1.0 M methylglyoxal + 1.0 M ammonium sulfate, 10 minutes after mixing, approx pH 7

**Figure S6.** Methylglyoxal (1.0 M) and ammonium sulfate (1.0 M) reactions, adjusted to pH 7 analyzed by APCI after 10 min using low temperature and low fragmentation conditions.

[Figure]

**Figure S7.** Methylglyoxal (1.0 M) and ammonium sulfate (1.0 M) reactions, adjusted to pH 9 analyzed by APCI after 10 min using low temperature and low fragmentation conditions. Methylglyoxal adjusted to pH 9 is shown in (b) as a control.

[Figure]

**Figure S8.** Methylglyoxal (1.0 M) prepared at a) pH 2, b) pH 5, c) pH 7, and d) pH 9 and allowed to dry uncovered for one week in the hood. The products observed in the reactions with AS do not appear in these samples.

[Figure]

**Figure S9.** Methylglyoxal (1.0 M) prepared at a) pH 2 and b) pH 9 and allowed to dry uncovered for one week in the hood (reproduced from Fig S8). In (c) and (d), duplicate capped samples are shown. The products observed in the reactions with AS do not appear in these samples.

[Figure]

**Figure S10.** Top: Total ion chromatogram for the ethyl acetate extract of a dried pH 2 sample. Middle: sum of all ions selected for SIM showing a peak at 4.175 min corresponding to the pyrazine internal standard and a second peak at 6.572 min corresponding to 2,5-dimethylpyrazine. Bottom: Electron impact spectrum acquired at retention time 6.560 (during 2,5-DMP elution) with prominent fragments at m/z 108, 42, and 81.

[Figure]

**Figure S11.** Top: spectrum obtained during elution of (proposed) 2,5-DMP peak from column. Bottom: NIST reference spectrum for electron impact ionization of 2,5-DMP.

[Figure]

**Figure S12.** Image taken after 24 hours of reaction time between methylglyoxal and AS in capped samples. A pH dependence on absorbance is immediately visible.

[Figure]

**Figure S13.** UV/visible absorbance spectra for methylglyoxal (1.0 M) and ammonium sulfate (1.0 M) at pH 5 under a variety of drying conditions.

[Figure]

**Figure S14.** Methylglyoxal (1.0 M) and ammonium sulfate (1.0 M), adjusted to pH 5 and evaporated to dryness over the course of one hour using HEPA-filtered laboratory air. The sample was analyzed a) immediately by APCI and b) 24 hours later when additional chromophores were observed.

[Figure]

**Figure S15.** Methylglyoxal (1.0 M) and ammonium sulfate (1.0 M), adjusted to pH 5 and evaporated to dryness over the course of one hour using ultra high purity nitrogen. The sample was analyzed a) immediately by APCI and b) 24 hours later when additional chromophores were observed.

[Figure]

**Figure S16.** Methylglyoxal (50 mM) and ammonium sulfate (50 mM), adjusted to pH 5 atomized and dried using diffusion dryers and impacted directly onto the APCI capillary. The sample was analyzed immediately by APCI.